# Improving the continuum of care monitoring in Brazilian HIV healthcare services: An implementation science approach

Ana Paula Loch[1][☉]*, Simone Queiroz Rocha[1][☉], Mylva Fonsi[1][☉], Joselita Maria de Magalhães Caraciolo[1][☉], Artur Olhovetchi Kalichman[1][‡], Rosa de Alencar Souza[1][‡], Maria Clara Gianna[1][‡], Alexandre Gonçalves[1][‡], Duncan Short[2][‡], Shenia Liane Pimenta[3][‡], Lea Bagnola[4][‡], Carolina Wonhnrath Menuzzo[5][‡], Zulmira da Rocha Meireles[6][‡], Eunice Natividade Diz[7][‡], Roberto Zajdenverg[8][‡], Isidoro Prudente[9][‡], Maria Ines Battistella Nemes[10][☉]

1 Centro de Referência e Treinamento DST/AIDS do Estado de São Paulo, São Paulo, Brazil, 2 ViiV Healthcare–Innovation and Implementation Science, London, England, 3 Grupo de Vigilância Epidemiológica de Campinas—GVE XVII—Secretaria Estadual de Saúde de São Paulo, Campinas, São Paulo, Brazil, 4 Programa Municipal de IST/Aids de Votuporanga, Votuporanga, São Paulo, Brazil, 5 Prefeitura Municipal de Sumaré, Sumaré, São Paulo, Brazil, 6 Grupo de Vigilância Epidemiológica de São José do Rio Preto, São Paulo, Brazil, 7 Grupo de Vigilância Epidemiológica de Osasco, Osasco, São Paulo, Brazil, 8 ViiV Healthcare–Scientific Affairs & Public Health, Rio de Janeiro, Brazil, 9 ViiV Healthcare–Head of Medical Affairs, Montevideo, Uruguay, 10 Departamento de Medicina Preventiva, Faculdade de Medicina da Universidade de Sao Paulo, Sao Paulo, estado de São Paulo, Brazil

☉ These authors contributed equally to this work.
‡ These authors also contributed equally to this work.
* anapaulaloch@gmail.com

**Data Availability Statement:** Data are held in a public repository at STI/AIDS State Program webpage (http://www.saude.sp.gov.br/centro-de-

## Abstract

### Objective

To evaluate the impact of an intervention improving the continuum of care monitoring (CCM) within HIV public healthcare services in São Paulo, Brazil, and implementing a clinical monitoring system. This system identified three patient groups prioritized for additional care engagement: (1) individuals diagnosed with HIV, but not receiving treatment (the treatment gap group); (2) individuals receiving treatment for >6 months with a detectable viral load (the virologic failure group); and (3) patients lost to follow-up (LTFU).

### Methods

The implementation strategies included three training sessions, covering system logistics, case discussions, and development of maintenance goals. These strategies were conducted within 30 HIV public healthcare services (May 2019 to April 2020). After each training session, professionals shared their experiences with CCM at regional meetings. Before and after the intervention, providers were invited to answer 23 items from the normalization process theory questionnaire (online) to understand contextual factors. The mean item scores were compared using the Mann–Whitney U test. The RE-AIM implementation science framework (evaluating reach, effectiveness, adoption, implementation, and maintenance) was used to evaluate the integration of the CCM.

referencia-e-treinamento-dstaids-sp/publicacoes/
publicacoes-download).

**Funding:** The study was supported by the ViiV Healthcare under Grant 210027 (https://viivhealthcare.com/en-gb/). DS, RZ and IP received salary from ViiV Health Care and GlaxoSmithKline. The funder provided support in the form of salaries for authors [DS, RZ and IP], but did not have any additional role in the study design, data collection and analysis, decision to publish, or preparation of the manuscript. The specific roles of these authors are articulated in the 'author contributions' section.

**Competing interests:** The commercial affiliation from authors DS, RZ and IP does not alter our adherence to PLOS ONE policies on sharing data and materials.

## Results

In the study, 47 (19.3%) of 243 patients with a treatment gap initiated treatment, 456 (49.1%) of 928 patients with virologic failure achieved suppression, and 700 of 1552 (45.1%) LTFU patients restarted treatment. Strategies for the search and reengagement of patients were developed and shared. Providers recognized the positive effects of CCM on their work and how it modified existing activities (3.7 vs. 4.4, p<0.0001, and 3.9 vs. 4.1, p<0.05); 27 (90%) centers developed plans to sustain routine CCM.

## Conclusion

Implementing CCM helped identify patients requiring more intensive attention. This intervention led to changes in providers' perceptions of CCM and care and management processes, which increased the number of patients engaged across the care continuum and improved outcomes.

## Introduction

The ongoing monitoring of people living with HIV/AIDS (PLHIV) is an essential component for achieving high levels of retention in care, antiretroviral therapy (ART) adherence, and viral suppression [1]. Brazilian national surveys have highlighted difficulties in terms of maintaining patient records and follow-up during every step of HIV care [2].

In Brazil, the Clinical Monitoring System (*Sistema de Monitoramento Clinico* [SIMC]) was developed in 2013 and made available by the Ministry of Health to all public healthcare services performing follow-up of PLHIV. The SIMC identifies three salient groups of patients for additional care engagement who have undergone CD4 and/or viral load tests: (1) those who have not started treatment (the treatment gap group), (2) individuals who have persistent detectable viral load ≥50 copies/mL after 6 months of ART initiation (the virologic failure group), and (3) patients who were lost to follow-up [3].

Patient listings are issued through interaction with the following databases: Laboratory Tests Control System (*Sistema de Controle de Exames Laboratoriais* [SISCEL]), which records tests of CD4 counts and viral load for every patient, and the Drug Logistic Control System (*Sistema de Controle Logístico de Medicamentos* [SICLOM]), which records the dispensation of antiretroviral drugs for every patient. The SIMC is routinely updated by the Ministry of Health, and listings are available to all public healthcare services managing PLHIV [3].

The issuance of listings allows care services to locate patients out of the assistance flow (or lost to the continuum of care) and develop actions and strategies to recruit patients not receiving treatment, analyze cases of virologic failure, and reengage those who have abandoned treatment. From a programmatic point of view, the evaluation of patients identified through the SIMC can contribute to the revision and improvement of assistance and management flow at local, regional, and municipal levels, and therefore, improve the quality of care provided by the healthcare services network. However, medical care practitioners and researchers have identified substantial difficulties in monitoring patients through the SIMC [2, 4].

The state of São Paulo accounts for approximately 25% of all patients receiving ART in Brazil (~150,000 of 594,000). Among diagnosed patients who had undergone a CD4 or viral load

test, 5.5% (10,296 patients) had a treatment gap, 5.9% (8,229 patients) had ART virologic failure, and 15.5% (27,611 patients) were lost to follow-up [5].

Since 2013, the STI/AIDS State Program has promoted initiatives for the improvement of HIV/AIDS care networks supported through epidemiological and service quality indicators. These initiatives include conducting regional workshops to develop action plans and goals. Managing patients effectively includes working with healthcare professionals from these health services as well as with regional and municipal managers responsible for the steps of HIV continuum of care, with regard to prevention, diagnosis, service referrals, maintenance of follow-up (i.e. retention), treatment adherence, and viral suppression [6].

A pilot intervention was conducted in 21 health services between 2018 and 2019, aiming to improve the implementation of continuum of care monitoring (CCM) with the SIMC within the health services. The intervention consisted of training in the use of the system and CCM conducted individually at each healthcare service. The intervention demonstrated strong adherence and effectiveness, promoting changes in assistance and management processes, and increasing the number of patients receiving optimal care across the care continuum [7].

The pilot intervention results motivated the development of a second intervention, which was conducted in three other health regions, comprising 33 healthcare services. The work presented here analyzes the effectiveness of the SIMC for CCM to identify and effectively treat patients who have not initiated treatment, have virologic failure, or are lost to follow-up.

## Materials and methods

### Study design

This was a hybrid type 3 mixed-method implementation study. Curran (2012) defines a hybrid type 3 as a study which "[tests an] implementation strategy while observing and gathering information on the clinical intervention's impact on relevant outcomes" [8]. This study tested an intervention to integrate the CCM within public services for PLHIV, gathering thorough and comprehensive information about the intervention with regard to the patients, professionals, and healthcare services. The intervention and intervention strategies present minimal risk. This intervention was strongly encouraged by the STI/AIDS Program and was adopted based on a successful pilot intervention conducted in a similar context [8]. The focus of this study included understanding the adoption and implementation of CCM, clinical impact of this intervention, and effects of changes in work processes.

Based on the Pragmatic Robust Implementation and Sustainability Model (PRISM) for translating research into practice [9], this conceptual framework was applied to identify which recipients could influence the CCM reach, adoption, implementation, maintenance, and effectiveness. Fig 1 shows the relevant contextual factors and the evaluation of intervention adoption and effectiveness according to the dimensions of the RE-AIM planning and evaluation framework [10].

The proportion of patients in the target population reached through this intervention was calculated using the following formula:

Treatment gap:

$$\frac{\text{Patients who were introduced to treatment} + \text{patients who refused treatment}}{\text{Patients meeting the treatment gap criteria}} \times 100$$

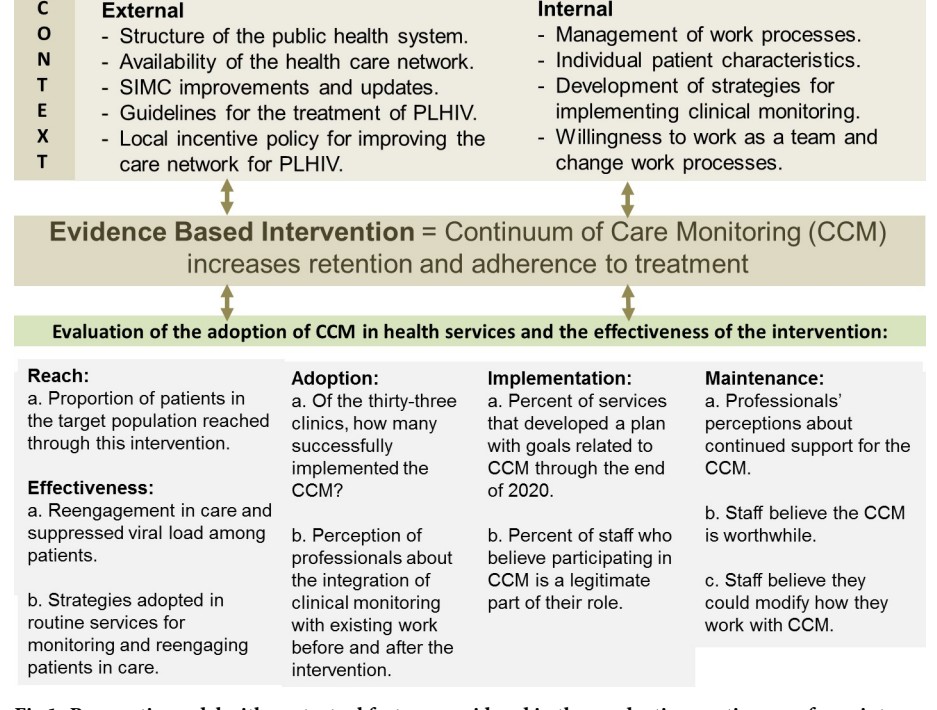

**Fig 1. Pragmatic model with contextual factors considered in the conducting continuum of care interventions and measurements.**

Virologic failure:

$$\frac{\text{Patients who received the intervention}}{\text{Patients who had virologic failure}} \times 100$$

Lost to follow-up:

$$\frac{\begin{array}{c}\text{Patients who restarted and remained on treatment at the end of the}\\ \text{intervention} + \text{patients who restarted and stopped treatment} +\\ \text{patients who started clinical monitoring, but did not start ART}\end{array}}{\text{Patients who were lost to follow} - \text{up.}} \times 100$$

## Intervention and strategy description

The process to improve the implementation of CCM with SIMC was conducted in three steps: pre-implementation, implementation, and post-implementation, as presented in Fig 2.

During pre-implementation, service professionals were invited to participate and answer the Normalization Measure Development (NoMAD) questionnaire to identify contextual factors impacting the implementation of the CCM. This questionnaire is based on the normalization process theory (NPT) and determines the collective behavior for the incorporation of complex interventions in practice. According to the NPT, the implementation of new practices in healthcare services is dynamic and dependent on the coordinated and collective behavior of individuals who work within the limits of healthcare contexts [11]. The NoMAD has 23 items distributed in four constructs: coherence, participation/cognitive engagement, collective action, and reflective monitoring [12].

**Objective of the intervention**
Improve implementation of patient clinical monitoring in the treatment gap, therapeutic failure and lost to follow-up groups.

**Pre-implementation**: Invitation to participate in the intervention, ethical procedures, and identification of hurdles, facilitators and professional perspectives involved in the implementation of continuum of care monitoring through the Normalization Measure Development (NoMAD) tool.

**I m p l e m e n t a t i o n**

**Phase 1**:
1.1 Technical visit for training on the use of the system, identification of users in the treatment gap group, therapeutic failure and lost to follow-up and cases discussion;
1.2 After approximately 60 days, regional meeting were conducted with presentation of progress of health services in monitoring cases identified in phase 1.1.

**Phase 2**:
2.1 Technical visit to identify new cases and monitor the situation of users identified in phase 1.1;
2.2 After approximately 60 days, a regional meeting was conducted for presenting progress of continuum of care monitoring of users included in phases 1.1 and 2.1.

**Phase 3**:
3.1 Technical visit to support development of action plans and goals related to continuum of care monitoring and closing of identified cases in phases 1.1 and 2.1;
3.2 After approximately 60 days, a regional meeting was conducted to present the final situation of users included in phases 1.1 e 2.1 as well as action plans and goals.

○ **May 2019**

○ **April 2020**

**Post-implementation**: evaluation of % of HIV/AIDS services that adopted routine clinical monitoring. Evaluation of achievement and effectiveness of intervention, implementation and maintenance of continuum of care monitoring after intervention. Normalization Measure Development (NoMAD) tool response.

**Fig 2. General structure of the intervention according to the steps of pre-implementation, implementation, and post-implementation of CCM.**

The following strategies that evolved from the Expert Recommendations for Implementing Change (ERIC) project were applied in this step: attainment of formal commitments from key partners with regard to the intervention, development of educational materials to support stakeholders in learning about the CCM, assessment of readiness and identification of barriers as well as facilitators with regard to the implementation of CCM, and the implementation of change by means of a leadership that declares CCM as a priority [13].

Each of the three implementation phases included an individual technical visit to each health service for approximately 4 h, followed by a 4-h face-to-face meeting within 60 days involving multiple services from the region (up to nine health services). The ERIC strategies [13] applied in phases 1, 2, and 3 involved the following tasks:

- Conducting educational outreach visits to train staff with regard to answering questions related to the use of the SIMC, and to train the team in the development of goals related to the CCM.

- Creating a SIMC training program for staff.

- Tailoring strategies to address barriers and leverage facilitators in terms of care flow.

- Gaining and sharing local knowledge of the CCM.

- Organizing implementation team meetings to support and provide health services with protected time to reflect on the CCM and share lessons learned.

- Promoting adaptability and identifying ways to tailor the CCM to meet local needs.

- Facilitating problem-solving related to the CCM.

- Carefully reexamining the

CCM implementation to monitor progress and adjust clinical practices.

All technical visits were conducted by four healthcare professionals trained in the STI/ AIDS program administered by the state of São Paulo. These professionals acted as technical support points, providing ongoing expert consultation with regard to strategies used to support CCM implementation as well as centralized technical assistance focused on CCM implementation issues for eight services participating in all implementation activities. The technical visit during the first CCM phase allowed for training of the medical care providers (physicians, nurses, pharmacists, social workers, psychologists) and administrative staff with regard to CCM. This technical visit identified and highlighted three circumstances through which patients could be lost to follow-up after diagnosis, as reported by the SIMC: treatment gap, virologic failure, and lost to follow-up, as presented in Fig 3.

A discussion of three real patient case studies identified by the SIMC report (for each site) was presented, covering each patient category described above. The group discussion presented patients' personal and clinical history data from medical records, and, when necessary, ART dispensation records. For each case, personalized strategies on how to contact users were developed within the group discussion, including direct contact or contact via other health institutions and social and/or home visits.

Within 60 days of the initial training, the participant sites met as a regional group to discuss selected CCM strategies and progress in terms of identifying and monitoring patients.

In the second phase, another technical visit was conducted at each site. During this visit, the local staff discussed their actions and follow-up related to the initial identified cases and identified new cases. The same topics were discussed during the second regional meeting.

In the third phase, the technical visit focused on supporting the health service to develop action plans and goals for the reduction of the number of patients with treatment gap, virologic failure, or lost to follow-up. A final regional meeting aimed to share presentations of action plans and goals related to the state plan across health services. However, due to COVID-19 emergence, this meeting was canceled, and the action plans and goals were shared with project coordination by the services through e-mail.

## Evaluation of the implementation of the CCM

The experiences and outcomes achieved through the implementation of the CCM were evaluated by considering:

- Cases identified.

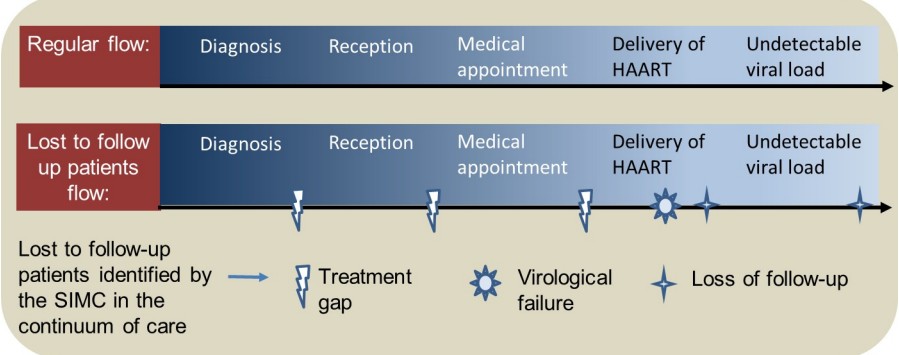

**Fig 3. The regular assistance flow and lost to follow-up flow of PLHIV in the continuum of care identified by the SIMC.**

- Strategies selected by the health services.

- Professionals' perspectives on the implementation.

**1. Individual monitoring of cases identified by the SIMC.** Patients who were followed up in the public healthcare system (*Sistema Único de Saúde*–[SUS]) were included in the monitoring. Patients who were listed twice in the systems, had death certificates in the official systems, had never been followed-up at the intervention healthcare service or transferred to other health services, were incorrectly registered as HIV-negative, and/or were followed-up in private services or the penitentiary were counted, but excluded from the monitoring.

For the virologic failure group, the following information was collected in the SISCEL and/or SICLOM databases: last viral load result (copies/mL), ART used by the patient, number of dispensed pills during the last 12 months, and the last detectable viral load and viral load status (detectable or undetectable) at the end of the intervention using SIM. The number of pills dispensed was divided by the number of expected days between the first and last dispensation dates. This result was multiplied by 100, which is the proportion of dispensed ART doses in the previous year and indicates the frequency of drug dispensation, an indirect indicator of treatment adherence.

We described patients' profiles in the lost to follow-up category and the time lost to follow-up. For patients returning to treatment during the intervention period, the proportion of dispensed drug doses was calculated during the period from re-initiation to the last dispensation and the end of study data collection to evaluate dispensation frequency and establish if the patient remained in follow-up or abandoned treatment again. Analyses were performed using standard software for statistics and data science (Stata/IC 14.0$^{\circledR}$).

**2. Identification of strategies contributing to patient monitoring in the routine services.** The identification of changes to assistance and management processes was obtained from the sites during regional meetings, which were held during phases 1.2 and 2.2 (Fig 2). Identified strategies were described according to their objectives and operationalization in routine care.

**3. Professionals' perspectives of the implementation.** Professionals' perspectives and subjective changes before and after implementation were obtained through responses to the NoMAD questionnaire [14, 15]. The questionnaire, which evaluated professionals' perception regarding the CCM, presents 20 items on a five-point Likert scale, ranging from "1 –strongly disagree" to "5 –strongly agree." The questionnaire was anonymously completed online. The means and standard deviations were calculated for each questionnaire item for the pre- and post-implementation phases, including all professionals who answered the questionnaire at these two time points.

For three items evaluating professionals' familiarity with the SIMC (0, still unfamiliar; 2–9; 10, strongly familiar), as well as current SIMC use and expectations of SIMC use (0, not at all; 1–4; 5, to a certain extent; 6–9; 10, definitely), the average calculation was performed using a 10-level Likert scale.

To evaluate statistically significant differences within professionals' perceptions regarding CCM implementation, responses to the items of the questionnaire (pre- and post-implementation) were compared as independent samples using the Mann–Whitney U test. Data were analyzed using Stata/IC 14.0$^{\circledR}$ software and are publicly available at http://www.saude.sp.gov.br/centro-de-referencia-e-treinamento-dstaids-sp/publicacoes/publicacoes-download.

## Population

The state of São Paulo comprises 645 cities clustered in 63 health regions and a public network of 198 specialized assistance services for the follow-up of approximately 150,000 PLHIV.

Thirty-three health services (16.6% of 198 in São Paulo State) from three health regions were invited to participate in the intervention, as presented in Fig 4.

The services were selected based on the volume of patients lost from the CCM in April 2019, before the intervention began, consisting of 1186 patients with treatment gaps (11.5% of the PLHIV diagnosed in the state), 1328 patients with virologic failure (14.8% of the PLHIV treated through state services), and 3449 patients who were lost to follow-up (12.5% of PLHIV with HAART prescribed through state services) [3]. The population monitored by healthcare services increased during the monitoring period due because of the addition of new cases in phase 2, as presented in Fig 2.

### Ethics statement

The study procedures were approved by the Ethical Committee of the University of São Paulo (protocol number: 3.270.762). Written consent was provided by each healthcare service involved prior to the intervention, as required by the local ethics committee. The study protocol was published in Registro Brasileiro de Ensaios Clínicos (U1111-1224-4363; http://www.ensaiosclinicos.gov.br/rg/RBR-9xrv64/).

## Results

Thirty services (90.9% of the 33 invited health services) from three health regions participated in the implementation (Table 1).

## Patient reports

Although the SIMC report identified a total of 1477 patients with treatment gaps, 1570 patients with virologic failure, and 3819 patients lost to follow-up, some of these patients did not meet the inclusion criteria, specified above, for these CCM intervention reports. After the healthcare

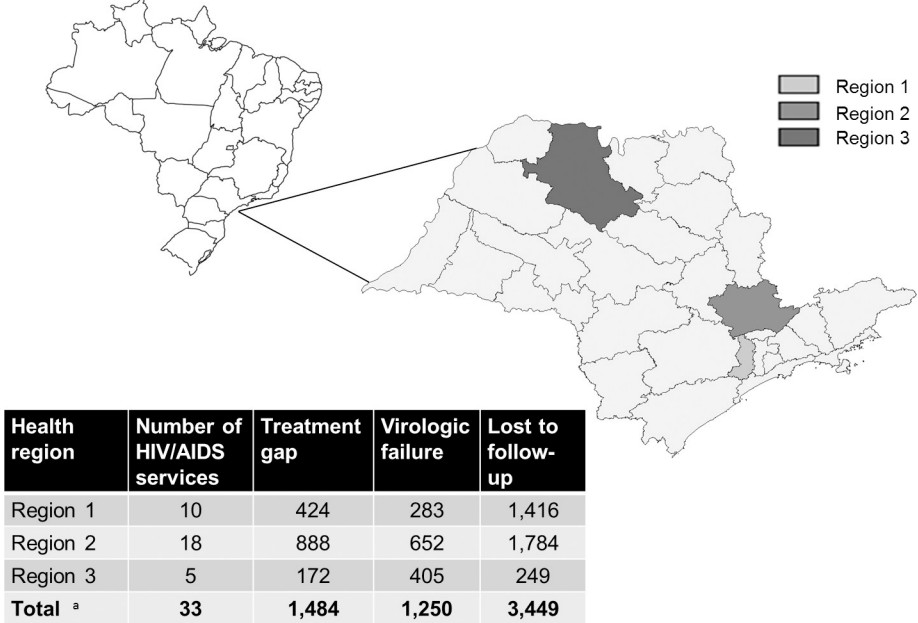

| Health region | Number of HIV/AIDS services | Treatment gap | Virologic failure | Lost to follow-up |
|---|---|---|---|---|
| Region 1 | 10 | 424 | 283 | 1,416 |
| Region 2 | 18 | 888 | 652 | 1,784 |
| Region 3 | 5 | 172 | 405 | 249 |
| Total [a] | 33 | 1,484 | 1,250 | 3,449 |

**Fig 4. Health regions and services included in the intervention and number of patients with treatment gap, virologic failure, or who were lost to follow-up.** [a] Clinical Monitoring System, May 2019.

**Table 1. Number of HIV healthcare services that participated in the intervention according to intervention phase and health region.**

| Health region | Number of HIV healthcare services | | Phase 1: System training and identification of PLHIV | | Phase 2: Clinical monitoring of PLHIV in three case situations | | Phase 3: Delivery of the final plan with goals regarding the CCM | |
|---|---|---|---|---|---|---|---|---|
| | Total invited | Accepted n (%) | Phase 1.1 | Phase 1.2 | Phase 2.1 | Phase 2.2 | Phase 3.1 | Phase 3.2 |
| **Region 1** | 10 | 9 (90) | 9 | 9 | 9 | 8 | 9 | 9 |
| **Region 2** | 18 | 16 (88.9) | 16 | 12 | 16 | 7 | 16 | 13 |
| **Region 3** | 5 | 5 (100) | 5 | 5 | 5 | 4 | 5 | 5 |
| **Total n (%)** | 33 | 30 (90.9) | 30 (100) | 26 (86.7) | 30 (100) | 19 (63.3) | 30 (100) | 27 (90%) |

CCM: continuum of care monitoring; HIV: human immunodeficiency virus; PLHIV: people living with HIV

Phase 3.2 was scheduled for the last week of March 2020, but was canceled due to the COVID-19 pandemic. A total of 27 services (90%) sent plans and goals via e-mail. Plans and goals were available in an online document on the STI/AIDS State Program webpage.

services reviewed the cases at the beginning of the intervention, only 278 (18.8%) were eligible for the treatment gap criteria, 1188 (75.7%) were eligible for the virologic failure criteria, and 1968 (51.7%) were eligible for the lost to follow-up criteria (Table 2).

## Treatment gap

At the end of the intervention, among 243 patients with treatment gap, 47 (19.3%) had started treatment, and 196 (80.7%) remained in the treatment gap category. Among these 196 patients, 86 (43.9%) were not located because of outdated telephone contact information, 3 (1.5%) did not provide consent for contact, 47 (24%) refused treatment, and 60 (30.6%) did not receive ART.

## Virologic failure

Among 1188 patients with virologic failure, 928 (78.1%) were reached by the intervention, of whom 456 (49.1%) presented viral suppression by the end of the intervention and 472 (50.9%) remained with detectable viral load, as presented in Fig 5. The other 260 patients were not

**Table 2. Patients' statuses in the SIMC report after analyses by the healthcare services.**

| Status | Treatment Gap | Virologic Failure | Lost to Follow Up |
|---|---|---|---|
| Death | 98 | 29 | 1126 |
| Duplicate records[a] | 955 [b] | 60 [c] | 236 |
| Transferred | 13 | 98 | 253 |
| No medical record in the healthcare service | 24 | 44 | 198 |
| HIV negative | 144 | 1 | 38 |
| VL suppression before review | NA | 150 | NA |
| **Excluded (sum of the above categories)** | **1234 (81.2%)** | **382 (24.3%)** | **1851 (48.3%)** |
| **Included/eligible** | **243 (18.8%)** | **1188 (75.7%)** | **1968 (51.7%)** |
| **Total** | **1477 (100%)** | **1570 (100%)** | **3819 (100%)** |

HIV: human immunodeficiency virus; LTFU: lost to follow up; SIMC: *Sistema de Monitoramento Clinico*; VL:

[a]Patients who had restarted treatment, but had duplicate records in the HAART system.

[b]Patients who had started ART, but appeared in the report due to duplicate records not initially identified through the interaction of databases for laboratory examinations and drug dispensation.

[c]Patients who had viral load suppression, but had duplicate records in the laboratory system.

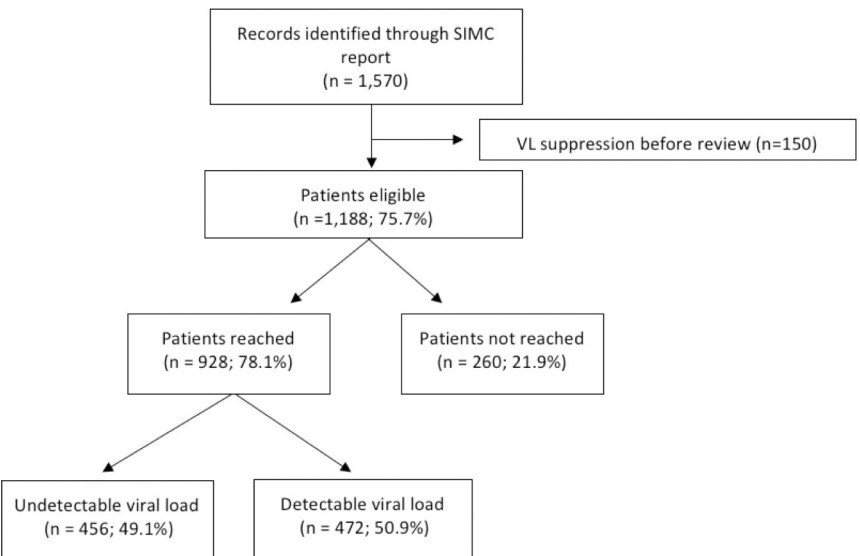

**Fig 5. Flow diagram of patients with virologic failure during the intervention.**

approached by the service during the intervention period, representing 21.9% of unreached patients.

Table 3 shows patients' characteristics (in the virologic failure group) at the beginning of the intervention. Most of these patients presented with a viral load of >500 copies/mL (52.2%). In terms of ART dispensation, 623 (52.5%) patients received <80% of the expected ART doses

**Table 3. Characteristics of patients with virologic failure (n = 1188).**

| Characteristics | N | % |
|---|---|---|
| **Last viral load** | | |
| <200 copies | 428 | 36 |
| 200_500 copies | 140 | 11.8 |
| >500 copies | 620 | 52.2 |
| **Percent of pills dispensed within 12 months before a detectable viral load** | | |
| Not applicable, patient returned from lost to follow-up | 102 | 8.6 |
| Not applicable, patient did not undergo viral load examination after starting ART | 56 | 4.7 |
| <60% | 344 | 29 |
| 60–80% | 279 | 23.5 |
| >80% | 407 | 34.3 |
| **Antiretroviral classes recorded in the patient history in the medication system after 2011** | | |
| NRTI (t)[a] + PI/r[b] | 309 | 26 |
| NRTI (t) + NNRTI[c] | 211 | 17.8 |
| NRTI (t) + INSTI[d] | 34 | 2.9 |
| Three or more classes of ART | 634 | 53.3 |

ART: antiretroviral therapy

[a]Nucleoside reverse-transcriptase inhibitor;

[b]protease inhibitors;

[c]non-nucleoside reverse-transcriptase inhibitors;

[d]integrase inhibitors

during the year preceding a detectable viral load, and 407 (34.3%) received >80% of the expected ART doses. A total of 554 (46.7%) patients presented with records of therapies containing two antiretroviral drug classes, and 634 (53.3%) had already changed their initial therapy (and thus had three or more antiretroviral drug classes in their therapeutic history).

## Lost to follow-up

Among 1968 patients who were lost to follow-up, 700 (35.6%) restarted and continued treatment until the end of the intervention, 138 (7%) restarted treatment, but stopped before the end of the intervention, 27 (1.4%) reengaged in the follow-up, but did not restart ART, and 1103 (56%) did not return to the service.

Table 4 shows the characteristics of the 700 patients who restarted the treatment. Among 247 (35.3%) patients who restarted after service intervention, most were lost to follow-up within the last 6 months. In terms of dispensation, 110 (44.5%) patients received <80% of the expected ART dose at the end of the intervention (Table 4).

Among the 1,103 patients who did not restart therapy, 39 (3.5%) were successfully contacted by phone, but refused to reengage with ART; 509 (46.1%) had an outdated telephone contact; 139 (12.6%) remained lost to follow-up after the services used several strategies to reach the patient, such as home visits and/or contact with other health services; 281 (25.5%) did not receive any intervention because the services needed to reorganize healthcare flows during COVID-19, and 135 (12.2%) did not receive any intervention, with no reason provided in the patient form.

## Healthcare service reports

Table 5 summarizes the strategies adopted by the services to conduct CCM and indicates that patients require more attention to effectively conduct these interventions.

Table 6 summarizes patient search and reengagement strategies developed by the health services during the intervention.

**Table 4. Characteristics of patients who were lost to follow-up and restarted therapy (n = 700).**

| | Total (n = 700) | | Spontaneous re-initiation (n = 453) | | After service intervention (n = 247) | | |
|---|---|---|---|---|---|---|---|
| **Characteristics** | **N** | **%** | **N** | **%** | **N** | **%** | **p-value [a]** |
| **Time since loss to follow-up** | | | | | | | **<0.001** |
| <6 months | 379 | 54.2 | 264 | **58.3** | 115 | 46.6 | |
| 6 months to 1 year | 196 | 28 | 121 | 26.7 | 75 | 30.4 | |
| 1–3 years | 87 | 12.4 | 51 | 11.3 | 36 | 14.6 | |
| >3 years | 38 | 5.4 | 17 | 3.7 | 21 | **8.5** | |
| **% of dispensed pills after restart of ART to the end of intervention** | | | | | | | **<0.01** |
| <60% | 212 | 30.3 | 151 | **33.3** | 61 | 24.7 | |
| 60–80% | 135 | 19.3 | 86 | 19 | 49 | 19.8 | |
| >80% | 248 | 35.4 | 163 | 36 | 85 | 34.4 | |
| Patients with only one ART dispensation | 105 | 15 | 53 | 11.7 | 52 | 21.1 | |

ART: antiretroviral therapy

[a]Chi-square test

**Table 5. Strategies tailored for health care professionals to identify PLHIV with a detectable viral load, treatment gap, and lost to follow-up (LFU).**

| Number of services | Strategy | Objective | Routine operationalization |
|---|---|---|---|
| 3 | Color scheme in medical records [a]. | Assist service professionals to identify immediately patients with a detectable viral load, treatment gap, and loss to follow-up. | Patients' medical records identified by the SIMC were classified by color scheme. Staff was capacitated to color meaning, and a guideline was made available to them. |
| 3 | Spreadsheet with patients missing their routine medical appointments [a]. | Monitor patients missing their medical appointments to prevent loss to follow-up. | At the end of the day, patients missing their appointment were contacted to schedule a new one. |
| 1 | Spreadsheet monitoring, medical appointments, ART dispensation, and viral load tests [b]. | Monitor patients in follow-up at the site. | Daily record of last viral load examination, last medical appointment, and ART dispensation in an Excel spreadsheet allows the service to classify patients as "active," "missing," or "loss to follow-up" and treatment and patients with laboratory examinations "delayed" or not. |
| 1 | Reference team in clinical monitoring [a]. | Establish a reference team in clinical monitoring in the service. | Team discusses cases available in the SIMC, exchanges experience in searching patients, and shares difficulties faced in performing their job. |
| 1 | Alert in electronic medical records used in all healthcare systems in the city (under implementation). | Increase the possibility of contact with patients with treatment gap and loss to follow-up. | Service evaluated as important the inclusion of an alert in the electronic medical record used by city healthcare services, asking patient to attend service to continue their treatment. |

[a] Strategy tailored during the intervention.

[b] Strategy tailored before the intervention and shared during group discussion.

## Professionals' report

The evaluation of CCM implementation from the perspective of participating professionals was conducted through the NoMAD questionnaire, which was answered by 129 professionals from 30 participant services during the pre-implementation phase; 45 professionals from 23 services returned the questionnaire during the post-implementation phase.

**Table 6. Strategies tailored by healthcare professionals to engage PLHIV with a detectable viral load, treatment gap, and/or LFU.**

| Number of services | Strategies | Objective | Routine operationalization |
|---|---|---|---|
| 4 | Home visits[a]. | Find patients who could not be contacted by phone. | After failing to contact patient by phone, service professionals performed home visit to understand the reasons for loss to follow-up and schedule a date to return to medical appointments. |
| 2 | Partnership with other services to search patients[a]. | Improve search processes of patients with loss to follow-up. | Service professionals of primary care and social workers were sensitized by HIV services to work together and search patients with loss to follow-up. |
| 1 | Training of pharmacists allocated in other locations[a]. | Include pharmacist in patient clinical monitoring in the service. | Pharmacists were capacitated to perform clinical monitoring aiming to reinforce the connection between the specialized service and dispensing pharmacy, allocated in other locations in the city. |
| 1 | Monitoring by the pharmacy patients with loss to follow-up[a]. | Establish a monitoring flow of patients with loss to follow-up by the service pharmacy. | Pharmacists became responsible to identify, contact, and record the outcomes of such actions in medical records to assist healthcare service team in clinical monitoring. |
| 2 | Availability of vacancy in the examination collection and/or medical appointment schedules[a]. | Prioritize examination collection and medical appointments of patients with virologic failure. | Service made available specific vacancy to attend patients identified in the SIMC as virologic failure in clinical appointments and viral load and CD4 examinations. |
| 1 | Nurse consultation after diagnosis[a]. | Connect the newly diagnosed patient to the service. | Newly diagnosed patients have appointment with a nurse to clarify their questions about the diagnosis and establish connection between the service and patient. |
| 1 | Preconsultation with a nurse[a]. | Improve clinical monitoring and patient retention. | Service implemented consultation with a nurse before routine medical appointment, to foster a space for their complaints and improve their retention in the service. |

[a] Strategy tailored during the intervention.

**Table 7. Professional perceptions on the use of continuum of care (CCM) monitoring in pre- and post-implementation steps.**

| CCM | Pre-implementation | | | Post-implementation | | | Mann–Whitney | |
|---|---|---|---|---|---|---|---|---|
| | N | Mean | DP | N | Mean | DP | p[a] | U test[b] |
| I am able to perceive how the CCM changes our current work routine. | 124 | 4.1 | 0.62 | 44 | 4.4 | 0.59 | <0.01 | 62.9 |
| The employees of this organization have a shared understanding of the purpose of the CCM. | 109 | 3.5 | 0.98 | 45 | 3.7 | 0.99 | 0.1527 | 56.8 |
| I understand how the CCM affects the essential activities of my own work. | 120 | 4.0 | 0.63 | 44 | 4.4 | 0.54 | <0.001 | 66.7 |
| I can see that CCM improves and facilitates my work. | 120 | 3.9 | 0.74 | 44 | 4.3 | 0.50 | <0.01 | 63.2 |
| There are people who decisively boost the use of the CCM and get others involved. | 117 | 3.9 | 0.81 | 45 | 3.9 | 0.74 | 0.7996 | 51.2 |
| I believe that participating in the CCM is a legitimate part of my role. | 125 | 4.0 | 0.71 | 45 | 4.4 | 0.57 | <0.01 | 64.2 |
| I am willing to take up new ways of working with colleagues, with regard to the CCM. | 126 | 4.2 | 0.54 | 45 | 4.3 | 0.69 | 0.2046 | 55.4 |
| I shall continue to provide my support for the CCM. | 122 | 4.2 | 0.54 | 45 | 4.4 | 0.58 | <0.05 | 60 |
| I can easily integrate the CCM into my existing work. | 118 | 3.6 | 0.85 | 45 | 4.0 | 0.82 | <0.001 | 65.4 |
| The CCM hinders labor relationships between workers. | 118 | 2.1 | 0.82 | 45 | 1.7 | 0.77 | <0.05 | 39 |
| I trust in the abilities of other people to use the CCM. | 120 | 3.8 | 0.74 | 44 | 3.9 | 0.83 | 0.4625 | 53.4 |
| The activities/functions related to the use of the CCM are given to professionals with adequate ability to perform them. | 119 | 3.9 | 0.63 | 45 | 4.1 | 0.67 | <0.05 | 60.8 |
| The staff receives sufficient training to enable them to implement the CCM. | 114 | 3.2 | 1.1 | 44 | 3.9 | 0.78 | <0.001 | 68.4 |
| The resources available are sufficient to give due support for the CCM. | 114 | 3.4 | 0.84 | 45 | 3.8 | 0.87 | <0.01 | 62.3 |
| The management gives appropriate support for the CCM. | 118 | 3.9 | 0.83 | 44 | 4.3 | 0.87 | <0.05 | 61.2 |
| I am aware of the reports made by professionals in health services regarding the impact of the use of the CCM. | 113 | 3.5 | 0.93 | 43 | 4.2 | 0.54 | <0.0001 | 70.2 |
| The employees at my health service agree that the CCM is worthwhile. | 117 | 3.9 | 0.74 | 45 | 4.1 | 0.76 | 0.1348 | 57 |
| I value the effects that the CCM has on my work. | 106 | 3.7 | 0.81 | 44 | 4.4 | 0.53 | <0.0001 | 72.8 |
| It is possible to use the team's feedback with regard to the SIMC to further improve the CCM in the future. | 113 | 3.9 | 0.74 | 45 | 4.3 | 0.66 | <0.001 | 64 |
| I am able to change my own way of working with the CCM. | 118 | 3.9 | 0.54 | 43 | 4.1 | 0.67 | <0.05 | 58.8 |

[a]Mann–Whitney U test.

[b]The proportion of comparisons in post-implementation responders presented higher values than in the pre-implementation step.

The familiarity of professionals with the CCM improved after implementation (with an average change from 1.6 at the start of the intervention to 7.2 at the end of the intervention, on a 10-point Likert scale; p<0.001). Most professionals also changed their perception regarding the use of CCM in the post-implementation phase (average change from 2.3 to 6.7 on a 10-point Likert scale; p<0.001).

No change was observed with regard to the professionals' future expectations of CCM implementation (average change from 7.1 to 7.8 on a 10-point Likert scale; p<0.1412).

After the intervention, the professionals had a better understanding of how CCM affected their work (with an average change of 4 vs. 4.4 on a five-point Likert scale; p<0.001) as well as the potential of CCM to affect and improve the work performed by professionals (with an average change 3.9 vs. 4.3 on a five-point Likert scale; p<0.01).

Table 7 summarizes the results from the NoMAD questionnaire, assessed using a five-point Likert scale, before and after the intervention. Regarding the adoption and maintenance of CCM, professionals started considering this intervention as a legitimate part of their work (average change 4 vs. 4.4; p<0.01), affirming that it could easily be integrated into their current activities (with an average change of 3.6 vs. 4.0; p<0.001), and reported that they would support CCM implementation in their service (with an average change of 4.2 vs. 4.4; p<0.05) (Table 7).

Service management support (with an average change of 3.9 vs. 4.3; p<0.05) and training (with an average change of 3.2 vs. 3.9; p<0.001) on the use of CCM also improved at the end

**Table 8. Intervention results according to the RE-AIM dimension.**

| RE-AIM | Indicator | Level | N (%) or Results |
|---|---|---|---|
| **REACH** | Patients with a treatment gap reached through this intervention | Patients | 94 (38.7%)[a] patients from the 243 identified. |
| | Patients with virologic failure reached through this intervention | Patients | 928 (78.1%)[b] patients from the 1,188 identified. |
| | Patients lost to follow-up reached through this intervention | Patients | 1,552 (78.6%)[c] patients from the 1,975 identified |
| **EFFECTIVENESS** | Patient with a treatment gap who were effectively introduced to the treatment | Patients | 47 (19.3%)[d] started treatment from the 243 reached. |
| | Patient with virologic failure who had suppressed viral load | Patients | 456 [e] (49.1%) patients had undetectable viral load from the 928 patients reached. |
| | Patient lost to follow-up who restarted treatment | Patients | 700[f] (45.1%) patients restarted treatment from the 1,552 reached |
| | Strategies adopted in routine services to monitor and reengage patients in care | Healthcare services | Improvements in healthcare service flows, described in Tables 5 and 6 |
| **ADOPTION** | Of the 33 clinics invited, how many successfully implemented clinical monitoring? | Healthcare professionals | 30 (90.9%) accepted from 33 invited |
| | Perception of professionals about the integration of clinical monitoring with the existing work | Healthcare professionals | 69 (53.5%) agreed or strongly agreed that they could easily integrate CCM before the intervention, and 37 (75.5%) agreed afterwards |
| **IMPLEMENTATION** | Percent of services that developed a plan with goals related to the CCM through the end of 2020 | Healthcare professionals | 27 (90.9%) of healthcare services |
| | Percentage of staff that believed that participating in the CCM was a legitimate part of their role | Healthcare professionals | 32 (24.8%) agreed or strongly agreed to participate in the CCM as part of their role before the intervention, and 43 (87.8%) afterwards |
| **MAINTENANCE** | Perception of professionals reporting continued support for the CCM | Healthcare professionals | 115 (89.1%) agreed or strongly agreed that they would continue to support the CCM before the intervention, and 43 (87.8%) agreed afterwards |
| | Staff that believed that the CCM was worthwhile | Healthcare professionals | 83 (64.3%) agreed or strongly agreed that CCM was worthwhile before the intervention, and 34 (69.4%) agreed afterwards |
| | Staff that believed that they could modify how they worked with the CCM | Healthcare professionals | 94 (72.9%) agreed or strongly agreed that they could modify their work before the intervention, and 39 (79.6%) agreed afterwards |

[a]Number of patients reached (47 started treatment + 47 refused treatment)/number of patients with treatment gap × 100.

[b] Number of patients reached/number of patients with virologic failure × 100.

[c] Number of patients reached (700 restarted and remained in treatment + 138 restarted then stopped treatment + 27 reengaged at follow-up + 39 refused + 139 remained on LFU after several strategies + 509 outdated telephone numbers)/number of patients in LFU × 100.

[d] Number of patients who started treatment through the end of the intervention/patients in the treatment gap × 100.

[e] Number of patients with undetectable viral load through the end of intervention/patients with virologic failure reached by intervention × 100.

[f] Number of patients who restarted treatment through the end of the intervention or patients who were lost to follow-up reached by the intervention × 100.

of the intervention. Professionals reported having gained knowledge about the effects of monitoring on their services (with an average change of 3.5 vs. 4.2; p<0.0001) and valued its impact on their activities (with an average change of 3.7 vs. 4.4; p<0.0001) (Table 7).

The results of the intervention are presented according to the dimensions of the RE-AIM implementation science framework, as shown in Table 8.

## Discussion

Implementation of the CCM contributed to improving the quality of care in the participating services with regard to four main aspects. These included (1) identifying PLHIV requiring more intensive care in the different phases of the care continuums, (2) introducing changes in the assistance and management processes, (3) fostering changes in the perception of professionals with regard to the relevance and need for monitoring, and (4) fostering the development and use of rates to establish programs and goals for quality improvement.

Most services were under-resourced with regard to staffing, which affected monitoring implementation. This barrier was also reported in a pilot study [7] and reflects the funding pressures within the public healthcare system in Brazil (SUS), which recently worsened due to constitutional amendments limiting funding [16]. Additionally, during the intervention period, the emergence of the COVID-19 pandemic impacted the organization of HIV services and canceled routine activities, maintaining only emergency care and antiretroviral dispensation within public health services, and some professionals were transferred to other services in the healthcare system [17, 18].

Despite the limited number of professionals, the services developed strategies to reorganize flows for searching and engaging patients in a timely manner. Strategies have been developed with low-cost tools and available resources, such as the development of spreadsheets for monitoring patients missing their appointments. Although widely regarded as a proxy for retention and adherence rates [19], routine monitoring of missing appointments was reported by only 35% of services attending PLHIV in Brazil [2, 20]. Additionally, changes in assistance flows required a more comprehensive clinical approach based on case discussions and the development of unique therapeutic projects [21].

The response to the NoMAD questionnaire showed that the implementation of monitoring caused relevant changes in the process of care, and that professionals intended to support its maintenance. However, there was a significant reduction in the number of professionals responding to the NoMAD during the post-implementation phase, indicating that monitoring was centered on specific members of the healthcare team, without complete involvement of available staff due to structural/organizational reasons.

Despite the systemic limitations with regard to availability of professionals, with consequent effects on workload and chronic management resourcing pressures, as well as the emergence of the pandemic, the intervention was effective even beyond the impact on the processes of care, resulting in the engagement of 19.3% of individuals without treatment, achieving viral suppression in 49.1% of patients who had virologic failure, and reengaging 45.1% of patients who were lost to follow-up. In addition, the system was able to apply an intervention that improved the quality of care offered in health services.

Ultimately, it is very likely that the effectiveness of the intervention was, in some cases, limited by patient characteristics. Patients identified as having "virologic failure" had treatment adherence issues, evaluated by the number of pills dispensed during the year before detectable viral load and the use of several antiretroviral classes. Half of the patients returning to health services after being lost to follow-up had medical records indicating irregular drug dispensation until the end of the intervention.

Loss of follow-up and virologic failure reflect adherence issues that can be associated with factors related to healthcare services and treatment [22–25]; however, they are also strongly determined by factors related to the patient [26], such as a lack of social and emotional support, which are usually not reverted by the action of primary healthcare services, particularly services with insufficient provision of medical specialties (such as psychiatry) and other sectors of social protection [2].

The implementation of a systemic intervention is a complex process that depends on several interactive and coexisting factors in healthcare, including adaptation to changes, availability of resources, organizational culture, management participation, definition of a clear implementation plan, monitoring, and feedback [27].

The intervention was able to address these factors and encourage the incorporation of CCM in routine service activities. The conclusion of the intervention, although hampered by the outbreak of the COVID-19 epidemic, was marked by 90% of the participating services enunciating clear goals for monitoring the continuum of care in an integrated manner as well

as the necessary changes in the service organization to achieve this. The goals developed by the services were organized in an official document within the state's STI/AIDS program.

## Limitations

This intervention faced an unavoidable limitation that was characteristic of the SIMC report. Based on the deterministic relationship between the medical examination and medication dispensing databases, we could not detect duplicates with sufficient sensitivity, resulting in a high number of patients mistakenly listed within various listings, forcing the local service to remove these patients from the intervention.

The study was not designed to compare our intervention results with results from similar control groups. Unfortunately, it is impossible to return to the SIMC to compare the results achieved in the regions implementing the intervention with other regions and/or the whole state at the same time period. A control group, however, although not feasible regarding the changes in workflow and process at the service level, could be included with respect to the SIMC outcomes related to treatment gap, virologic failure, and loss of follow-up. In contrast, this intervention, supported by the STI/AIDS Program from the State of Sao Paulo, achieved important results in patients, healthcare services, and at the program level when comparing pre-and post-intervention outcomes. We hope our results will support interventions and improve on the reports available in the system.

The number of providers answering the NoMAD questionnaire at the end of the intervention decreased compared to the number answering the questionnaire at the start of the intervention. This could be related to two factors observed, although not measured, during the intervention: (1) at the beginning of the intervention, managers asked all healthcare service providers to answer the questionnaire, even before defining whether all professionals would participate in the intervention, and (2) the CCM activities were centered on specific members of each healthcare service (i.e. the majority of questionnaire respondents at the end of the intervention). In addition, the NoMAD questionnaire was answered anonymously to protect participant privacy, which limits the paired comparison.

The assessment of the proportion of ART doses dispensed to patients who returned from lost to follow-up provides a provisional evaluation of patient return. Evaluation of the impact of the intervention on the long term maintenance of these patients' treatment programs during follow-up must be considered in future evaluations.

## Acknowledgments

We would like to thank all health professionals from the regional epidemiological surveillance and healthcare services involved in this study.

## Author Contributions

**Conceptualization:** Ana Paula Loch, Simone Queiroz Rocha, Mylva Fonsi, Joselita Maria de Magalhães Caraciolo, Artur Olhovetchi Kalichman, Rosa de Alencar Souza, Maria Clara Gianna, Alexandre Gonçalves, Duncan Short, Roberto Zajdenverg, Isidoro Prudente, Maria Ines Battistella Nemes.

**Data curation:** Ana Paula Loch, Simone Queiroz Rocha, Mylva Fonsi, Shenia Liane Pimenta, Lea Bagnola, Carolina Wonhnrath Menuzzo.

**Formal analysis:** Ana Paula Loch.

**Funding acquisition:** Ana Paula Loch, Artur Olhovetchi Kalichman, Rosa de Alencar Souza, Maria Clara Gianna, Alexandre Gonçalves, Roberto Zajdenverg, Isidoro Prudente.

**Investigation:** Ana Paula Loch, Simone Queiroz Rocha, Mylva Fonsi, Joselita Maria de Magalhães Caraciolo, Shenia Liane Pimenta, Lea Bagnola, Carolina Wonhnrath Menuzzo, Zulmira da Rocha Meireles, Eunice Natividade Diz, Maria Ines Battistella Nemes.

**Methodology:** Ana Paula Loch, Simone Queiroz Rocha, Mylva Fonsi, Joselita Maria de Magalhães Caraciolo, Duncan Short.

**Project administration:** Ana Paula Loch, Maria Ines Battistella Nemes.

**Resources:** Ana Paula Loch.

**Software:** Ana Paula Loch.

**Supervision:** Ana Paula Loch, Simone Queiroz Rocha, Mylva Fonsi, Joselita Maria de Magalhães Caraciolo, Maria Ines Battistella Nemes.

**Validation:** Ana Paula Loch, Shenia Liane Pimenta, Lea Bagnola, Carolina Wonhnrath Menuzzo.

**Visualization:** Ana Paula Loch.

**Writing – original draft:** Ana Paula Loch, Simone Queiroz Rocha, Mylva Fonsi, Joselita Maria de Magalhães Caraciolo, Artur Olhovetchi Kalichman, Rosa de Alencar Souza, Maria Clara Gianna, Alexandre Gonçalves, Duncan Short, Maria Ines Battistella Nemes.

**Writing – review & editing:** Ana Paula Loch, Simone Queiroz Rocha, Mylva Fonsi, Joselita Maria de Magalhães Caraciolo, Artur Olhovetchi Kalichman, Rosa de Alencar Souza, Maria Clara Gianna, Alexandre Gonçalves, Duncan Short, Shenia Liane Pimenta, Lea Bagnola, Carolina Wonhnrath Menuzzo, Zulmira da Rocha Meireles, Eunice Natividade Diz, Roberto Zajdenverg, Isidoro Prudente, Maria Ines Battistella Nemes.

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
