## [Decision Letter · Decision Letter 0]

7 Nov 2020

PONE-D-20-23813

Improving the Continuum of Care Monitoring in Brazilian HIV Health Care Services: An implementation science approach

PLOS ONE

Dear Dr. Loch,

Thank you for submitting your manuscript to PLOS ONE. After careful consideration, we feel that it has merit but does not fully meet PLOS ONE’s publication criteria as it currently stands. Therefore, we invite you to submit a revised version of the manuscript that addresses the points raised during the review process.

Both reviewers highlight important methodological and presentation issues with the current manuscript that need to be addressed carefully in a revision.

We look forward to receiving your revised manuscript.

Kind regards,

Bruce A Larson

Academic Editor

PLOS ONE

Journal Requirements:

3. Thank you for including your ethics statement: 

"All procedures performed in studies involving human participants were in  accordance  with  the  ethical  standards  of  the  institutional  and/or  national research  committee  (Ethical  Committee of  the  University  of  São  Paulo -protocol  number: 3.270.762)and  with  the  1964  Helsinki  declaration  and  its later  amendments  or  comparable  ethical  standards. The  study  protocol  was published  in  Brazilian  registry  of  clinical  research  (UTN  Number:  U1111-1224-4363 -http://www.ensaiosclinicos.gov.br/rg/RBR-9xrv64/)."

a. Please amend your current ethics statement to confirm that your named institutional review board or ethics committee specifically approved this study.

4.Thank you for providing the following Funding Statement: 

'The study was supported by the ViiV Healthcare under Grant 210027 (https://viivhealthcare.com/en-gb/). DS, RZ and IP received salary from ViiV Health Care and GlaxoSmithKline. '

We note that one or more of the authors is affiliated with the funding organizations, indicating the funder may have had some role in the design, data collection, analysis or preparation of your manuscript for publication; in other words, the funder played an indirect role through the participation of the co-authors.

a. If the funding organization did not play a role in the study design, data collection and analysis, decision to publish, or preparation of the manuscript and only provided financial support in the form of authors' salaries and/or research materials, please review your statements relating to the author contributions, and ensure you have specifically and accurately indicated the role(s) that these authors had in your study in the Author Contributions section of the online submission form. Please make any necessary amendments directly within this section of the online submission form. 

Please also update your Funding Statement to include the following statement: “The funder provided support in the form of salaries for authors [insert relevant initials], but did not have any additional role in the study design, data collection and analysis, decision to publish, or preparation of the manuscript. The specific roles of these authors are articulated in the ‘author contributions’ section.”

If the funding organization did have an additional role, please state and explain that role within your Funding Statement.

Additional Editor Comments:

Both Reviewers provide thoughtful comments and questions in their reviewers, but arrived at different recommendations.

You are invited to revise and resubmit if you carefully address each comment/question in each reviewers report.

Reviewers' comments:

Reviewer's Responses to Questions

**Comments to the Author**

1. Is the manuscript technically sound, and do the data support the conclusions?

Reviewer #1: Partly

Reviewer #2: Partly

2. Has the statistical analysis been performed appropriately and rigorously? 

Reviewer #1: I Don't Know

Reviewer #2: No

3. Have the authors made all data underlying the findings in their manuscript fully available?

Reviewer #1: Yes

Reviewer #2: Yes

4. Is the manuscript presented in an intelligible fashion and written in standard English?

Reviewer #1: No

Reviewer #2: Yes

5. Review Comments to the Author

Reviewer #1: In general, the manuscript is technically sound. It reports on an implementation study to improve use of the Brazilian clinical monitoring system (SIMC) in service settings to identify gaps in the HIV care continuum and people with HIV (PWHIV) who are either in treatment gap, virologic failure, or lost to care. The study was designed to improve service organizations’ adoption of the use of SIMC to identify these cases, and to strategize ways to reengage them in care and improve treatment outcomes. The evaluation design assessed both the training and technical support process with service providers/staff and the health outcomes of PWHIV identified as out of care or not in virologic suppression. The study is particularly significant given the value globally of using nationally and locally available data monitoring systems to identify gaps in the HIV care continuum system to achieve significant improvements in population level health outcomes.

The study used a “hybrid type 3 mixed-method implementation study” design. However, this is not well defined when introduced, and the reader is left to infer what that means.

The study used well-accepted theoretical (REAIM) and methodological (PRISM) approaches to implementation research and evaluation in real world settings. The intervention process was well described and well-staged to improve implementation success, with repeated training and follow-up and multiple time point evaluation and outcome measures.

Unfortunately, Figure 2 needs revision, because it contains too much text about the implementation design in font sizes and colors that (even in the TIFF file) are not sufficiently legible. This level of detail should be laid out in the text of the manuscript, leaving only the design outlines for the figure. Likewise, Figure 5 on the REAIM component findings is difficult to read, and would be better presented in the text of the manuscript as numbered or bulleted points below each of the REAIM constructs.

The multilevel evaluation design is strong, including using the national electronic monitoring system (SIMC) to measure improvements in patient outcomes, the repeated listening sessions to document provider perspectives on using the system, documentation of strategies and service plans for reaching patients needing follow-up, and plans for sustaining use of the monitoring system. Additionally, a pre/post survey (NoMAD) was used with providers at the 30 participating service organizations to measure their perspectives on adopting routine activities to incorporate and sustain use of the clinical monitoring system as part of their routine HIV care continuum activities. However, a better description of this questionnaire would be beneficial up front, or at least reference to Table 7 for full content of the survey.

The methods for calibrating treatment gaps, virologic failure and lost to care using the SIMC appear reasonable and rigorous, including steps to remove duplicates and ineligible patients from the list of target patients. However, the presentation in Table 2 is difficult to follow in terms of conditions of the patients in the database before the intervention and/or before/after sorting for targeted services.

Findings related to patient/client health appear to be robust and reliable, given that they are derived from the ongoing national monitoring system from which patients needing extra services were identified. Also, strategies both to indicate and to engage PLHIV who are in treatment gap, virologic failure, or lost to care are well described. However, there are two Tables 6 presenting these data, which should be differently numbered.

While the study was strong in relation to replication of a rigorous intervention implementation and evaluation design systematically conducted in 30 different service settings, it might have been strengthened and the conclusions more strongly supported by comparing patient outcomes in similar comparison communities and/or service organizations that did not receive the intervention training and support program, in addition to those that did. This limitation to the study design should at least be mentioned in the Limitations section.

The pre/post analyses of responses on the NoMAD survey need further analysis. While the comparison statistics seem appropriate, the very high attrition rate (65%) raises the question of attrition bias in the follow-up sample and challenges confidence in the strength of the findings. This should be addressed by explaining better who did not respond to the follow-up and possible reasons they did not. The current comment, that the non-responders are likely to be service staff not responsible for these tasks, is insufficient. Additionally, this low response rate to the follow-up NoMAD survey should be mentioned in the Limitations section.

The protocol has been made available through the Brazilian clinical research registry. However, it is not clear if all data are available to the public.

The whole manuscript, including Figures, would benefit from additional proofreading for grammatical and usage issues or missing words. Particular challenges with clarity include the last sentence before Table 6, and sections of the Discussion, especially the 4th, 5th, and 6th paragraphs, among other places here and there throughout the manuscript.

Reviewer #2: Thank you for the opportunity to review this manuscript. The authors present an evaluation of a technical support strategy to increase uptake of clinical monitoring of patients with HIV along the treatment cascade in Sao Paulo, Brazil. They used a mixed methods approach, combining pre/post questionnaires with providers, reviews of clinical databases to track patients through the care cascade, and notation of strategies adopted by providers to support uptake of CCM. Questionnaire results suggest that providers are enthusiastic about CCM and plan to sustain it. However, the patient-level analysis is not informative as to the effects of the implementation strategy, as there are no comparators (either pre-implementation or control) against which to judge relative uptake of CCM and its subsequent effects. The paper contains many useful pieces of information but is generally dense and the findings are difficult to interpret. I would recommend the authors consider a quasi-experimental evaluation approach to judge the effects of the technical support implementation strategy on CCM uptake and patient outcomes (e.g., tracking rates of virologic failure among implementing facilities pre- and post-implementation), or reframe this paper as a case study. See below for some major critiques:

- Perhaps most importantly, it is not clear what the data in Tables 2-6 tell us about the effects of the technical support implementation strategy on CCM usage or patient-level cascade outcomes. Table 6 comes closest by comparing patients pre- and post-implementation, but here the focus is on comparing patient characteristics, not patient probability of restarting therapy conditional on being in the pre- or post-implementation period. Useful patient-level estimates of effectiveness would answer questions like, “What proportion of clinic patients are in virologic failure pre-implementation, vs. what proportion are in virologic failure post-implementation?” Ideally, comparisons like this would also include a pre-post comparison with one or more control facilities.

- It is unclear how the PRISM framework was incorporated.

- Several of the RE-AIM components are inappropriately mapped to study indicators. For example, Reach implies an estimate of the number of patients served divided by the number of eligible patients. In this study, your reach indicators appear to be the number of patients still remaining in each cascade bucket (e.g., virologic failure) after data cleaning.

- Figure 2 makes it appear that the NoMAD questionnaire was only given at pre-implementation, but Methods Section 3 implies that NoMAD was given pre- and post-implementation to track changes in provider perspectives. Please clarify in Figure 2.

- Regarding the pre/post CCM implementation questionnaire analysis – do you treat the pre- and post- observations as independent, or do you account for observations of the same provider at pre- and post (paired analysis)? This is not clear.

- Methods/Population section – were facilities randomly sampled on the basis of patient volume lost to care, or purposively selected?

6. PLOS authors have the option to publish the peer review history of their article (what does this mean?). If published, this will include your full peer review and any attached files.

Reviewer #1: No

Reviewer #2: No

---

## [Author Response · Author response to Decision Letter 0]

5 Jan 2021

Editors’ comments: 

Author’s answer: The manuscript was carefully reviewed according PLOS ONE's style requirements. 

Author’s answer: The manuscript was reviewed by EDITAGE and submitted using tracking changes as supporting information file.

3. Thank you for including your ethics statement: 

"All procedures performed in studies involving human participants were in accordance with the ethical standards of the institutional and/or national research committee (Ethical Committee of the University of São Paulo -protocol number: 3.270.762) and with the 1964 Helsinki declaration and its later amendments or comparable ethical standards. The study protocol was published in Brazilian registry of clinical research (UTN Number: U1111-1224-4363 - http://www.ensaiosclinicos.gov.br/rg/RBR-9xrv64/)."

a. Please amend your current ethics statement to confirm that your named institutional review board or ethics committee specifically approved this study.

 Author’s answer: The IRB name was confirmed and replaced in the sections recommended. 

4.Thank you for providing the following Funding Statement: 

'The study was supported by the ViiV Healthcare under Grant 210027 (https://viivhealthcare.com/en-gb/). DS, RZ and IP received salary from ViiV Health Care and GlaxoSmithKline. '

We note that one or more of the authors is affiliated with the funding organizations, indicating the funder may have had some role in the design, data collection, analysis or preparation of your manuscript for publication; in other words, the funder played an indirect role through the participation of the co-authors.

a. If the funding organization did not play a role in the study design, data collection and analysis, decision to publish, or preparation of the manuscript and only provided financial support in the form of authors' salaries and/or research materials, please review your statements relating to the author contributions, and ensure you have specifically and accurately indicated the role(s) that these authors had in your study in the Author Contributions section of the online submission form. Please make any necessary amendments directly within this section of the online submission form. 

Please also update your Funding Statement to include the following statement: “The funder provided support in the form of salaries for authors [insert relevant initials], but did not have any additional role in the study design, data collection and analysis, decision to publish, or preparation of the manuscript. The specific roles of these authors are articulated in the ‘author contributions’ section.”

If the funding organization did have an additional role, please state and explain that role within your Funding Statement.

Author’s answer: The authors from funding organization played a role in the study design, decision to publish and or preparation of the manuscript. The author’s role was updated in the online submission form. 

Author’s answer: As recommended, the competing Interests Statement was added on the manuscript. Changes are available bellow: 

Competing Interests 

The commercial affiliation from authors DS, RZ and IP does not alter our adherence to PLOS ONE policies on sharing data and materials.

 

Reviewers' comments:

Reviewer #1: In general, the manuscript is technically sound. It reports on an implementation study to improve use of the Brazilian clinical monitoring system (SIMC) in service settings to identify gaps in the HIV care continuum and people with HIV (PWHIV) who are either in treatment gap, virologic failure, or lost to care. The study was designed to improve service organizations’ adoption of the use of SIMC to identify these cases, and to strategize ways to reengage them in care and improve treatment outcomes. The evaluation design assessed both the training and technical support process with service providers/staff and the health outcomes of PWHIV identified as out of care or not in virologic suppression. The study is particularly significant given the value globally of using nationally and locally available data monitoring systems to identify gaps in the HIV care continuum system to achieve significant improvements in population level health outcomes.

The study used a “hybrid type 3 mixed-method implementation study” design. However, this is not well defined when introduced, and the reader is left to infer what that means.

Author’s answer: As recommended, the hybrid type 3 mixed-method implementation study concept was incorporated when introduced in the section Study design. Changes are available bellow: 

This is a hybrid type 3 mixed-method implementation study that tests an intervention to integrate the CCM within public services for PLHIV, gathering information on the intervention and related outcomes. The intervention presents minimal risk associated with the strategies used. It was strongly encouraged by the STI/AIDS Program and adopted from a pilot intervention in a similar context setting [8]. The focus includes understanding the adoption of CCM, clinical impact, and changes in work processes. 

The study used well-accepted theoretical (REAIM) and methodological (PRISM) approaches to implementation research and evaluation in real world settings. The intervention process was well described and well-staged to improve implementation success, with repeated training and follow-up and multiple time point evaluation and outcome measures. Unfortunately, Figure 2 needs revision, because it contains too much text about the implementation design in font sizes and colors that (even in the TIFF file) are not sufficiently legible. This level of detail should be laid out in the text of the manuscript, leaving only the design outlines for the figure. Likewise, Figure 5 on the REAIM component findings is difficult to read, and would be better presented in the text of the manuscript as numbered or bulleted points below each of the REAIM constructs.

Author’s answer: - As recommended, Figure 2 was revised and the text about strategies applied was reallocated in the text of the manuscript. Changes are available bellow: 

Each of the three implementation phases included an individual technical visit to each service for approximately 4 h, followed by a 4-h face-to-face meeting within 60 days involving multiple services from the region (up to nine health services). ERIC strategies [13] applied in phases 1, 2, and 3 involved: 

- Conduct educational outreach visits to train staff to answer questions related to the use of SIMC and train the team in the development of goals related to the CCM.

- Make training dynamic involving staff in SIMC use.

- Tailor strategies to address barriers and leverage facilitators in care flow.

- Gain and share local knowledge on CCM.

- Organize implementation team meetings to support and provide them protected time to reflect about CCM and share lessons learned.

- Promote adaptability, identifying ways CCM can be tailored to meet local needs.

- Facilitate problem solving related to CCM. 

- Purposely reexamine the implementation to monitor progress and adjust clinical practices. 

Figure 5 was changed to Table 6, where we review and present the results for each REAIM constructs, improving the reading and understanding about how we achieve these results. Changes are available bellow: 

The results of the intervention are presented according to the dimension of the RE-AIM implementation science framework, as shown in Table 6. 

Table 6. Intervention results according to the RE-AIM dimension

RE-AIM Indicator Level N (%) or Results

REACH

 Patients with treatment gap reached through this intervention Patients 94 (38.7%)a patients from 243 identified.

 Patients with virologic failure reached through this intervention Patients 928 (78.1%)b patients from 1,188 identified.

 Patients with loss to follow-up reached through this intervention Patients 1,552 (78.6%)c patients from 1,975 identified

EFFECTIVENESS

 Patient with treatment gap who were introduced to the treatment Patients 47 (19.3%)d started treatment from 243 reached. 

 Patient with virologic failure who had suppressed viral load Patients 456 e (49.1%) patients had undetectable viral load from 928 patients reached. 

 Patient with loss of follow-up who restarted treatment Patients 700f (45.1%) patients restarted treatment from 1,552 reached

 Strategies adopted in routine of services to monitor and reengage patients in care Healthcare services Improvements in healthcare services flows, described in Figures 6 and 7

ADOPTION

 Of the 33 clinics invited, how many successfully implemented the clinical monitoring Healthcare professionals 30 (90.9%) accepted from 33 invited

 Perception of professionals about the integration of the clinical monitoring to the existing work Healthcare professionals 69 (53.5) agreed or strongly agreed that they could easily integrate CCM before the intervention and 37 (75.5%) afterward

IMPLEMENTATION Percent of services that developed the plan with goals related to CCM until the end of 2020 Healthcare professionals 27 (90.9%) healthcare services

 Percentage of staff that believed that participating in the CCM is a legitimate part of their role Healthcare professionals 32 (24.8%) agreed or strongly agreed to participate in CCM as part of their role before the intervention and 43 (87.8%) afterward

MAINTENANCE Perception of professionals about continued support for the CCM Healthcare professionals 115 (89.1%) agreed or strongly agreed that they will continue to support CCM before the intervention and 43 (87.8%) afterward

 Staff believed that the CCM is worthwhile Healthcare professionals 83 (64.3%) agreed or strongly agreed that CCM is worthwhile before the intervention and 34 (69.4%) afterward

 Staff believed that they may modify how they work with the CCM Healthcare professionals 94 (72.9%) agreed or strongly agreed that they may modify their work before the intervention and 39 (79.6%) afterward

aNumber of patients reached (47 started treatment + 47 refused treatment)/number of patients with treatment gap × 100. 

b Number of patients reached/number of patients with virologic failure × 100. 

c Number of patients reached (700 restarted and remained treatment + 138 restarted and stopped + 27 just reengaged follow-up+ 39 refused + 139 remained on LFU after several strategies + 509 outdated telephone numbers)/number of patients in LFU × 100.

d Number of patients started treatment until the end of intervention/patients in treatment gap × 100. 

e Number of patients with undetectable viral load until the end of intervention/patients with virologic failure reached by intervention × 100. 

f Number of patients who restarted treatment until the end of the intervention/patients with loss of follow-up reached by the intervention × 100. 

The multilevel evaluation design is strong, including using the national electronic monitoring system (SIMC) to measure improvements in patient outcomes, the repeated listening sessions to document provider perspectives on using the system, documentation of strategies and service plans for reaching patients needing follow-up, and plans for sustaining use of the monitoring system. Additionally, a pre/post survey (NoMAD) was used with providers at the 30 participating service organizations to measure their perspectives on adopting routine activities to incorporate and sustain use of the clinical monitoring system as part of their routine HIV care continuum activities. However, a better description of this questionnaire would be beneficial up front, or at least reference to Table 7 for full content of the survey.

Author’s answer: A better description of the questionnaire was incorporate in Materials and Methods section, adding more two references related to the questionnaire. The paper which presents the cross-cultural adaptation of the NoMAD questionnaire to Brazilian Portuguese was published in October and the reference number 15 was updated. Changes are available bellow: 

During the pre-implementation, service professionals were invited to participate and answer the Normalization Measure Development (NoMAD) questionnaire to identify contextual factors impacting the implementation of the CCM. This questionnaire is based on the normalization process theory (NPT) and determines the collective behavior for the incorporation of complex interventions in practice. According to the NPT, the implementation of new practices in healthcare services is dynamic and dependent on the coordinated and collective behavior of individuals who work within the limits of healthcare contexts [11]. NoMAD has 23 items distributed in four constructs: coherence, participation or cognitive engagement, collective action, and reflective monitoring [12].

The methods for calibrating treatment gaps, virologic failure and lost to care using the SIMC appear reasonable and rigorous, including steps to remove duplicates and ineligible patients from the list of target patients. However, the presentation in Table 2 is difficult to follow in terms of conditions of the patients in the database before the intervention and/or before/after sorting for targeted services.

Author’s answer: As recommended, Table 2 was improved to better understanding. We transform Tables 2, 3 and 5 in just on table (Table 2) because all of these Tables comprised ineligibles patient from different reports, but with similar situations (p.ex.: Death or duplicate records). Changes are available bellow: 

Table 2. Patients’ status in SIMC report after analyses by the healthcare services 

Status Gap (%) Failure (%) LTFU (%)

Death 98 (7.9) 29 (1.8) 1126 (60.8)

Duplicate records 955 (77.4)* 60 (3.8)# 236 (12.7)

Transferred 13 (1.1) 98 (6.2) 253 (13.7)

No medical record in the healthcare service 24 (1.9) 44 (2.6) 198 (10.7)

HIV negative 144 (11.7) 1 (0.1) 38 (2.1)

VL suppression before review NA 150 (9.6) NA

Total 1430 (100) 1570 (100) 3819 (100)

Excluded 1,234 (81.2) 382 (24.3) 1,851 (48.3)

Included/ eligible 243 (18.8) 1,188 (75.7) 1,968 (51.7)

 *Patients had started ART but appeared in the report due to duplicate records not identified by the interaction of databases for laboratory examinations and drug dispensation.

#Patients had viral load suppression but had duplicate records in the laboratory system.

^Patients had restarted treatment but had duplicate records in the HAART system.

Findings related to patient/client health appear to be robust and reliable, given that they are derived from the ongoing national monitoring system from which patients needing extra services were identified. Also, strategies both to indicate and to engage PLHIV who are in treatment gap, virologic failure, or lost to care are well described. However, there are two Tables 6 presenting these data, which should be differently numbered.

Author’s answer: As recommended, The Table/Figure number was corrected. 

While the study was strong in relation to replication of a rigorous intervention implementation and evaluation design systematically conducted in 30 different service settings, it might have been strengthened and the conclusions more strongly supported by comparing patient outcomes in similar comparison communities and/or service organizations that did not receive the intervention training and support program, in addition to those that did. This limitation to the study design should at least be mentioned in the Limitations section.

Author’s answer: The limitation section was updated to include to approach this limitation in design. Changes are available bellow: 

The study was not designed to compare the results with a similar control group. Unfortunately, it is impossible to return to the SIMC to compare the results achieved in the regions with intervention with other regions or the whole state at the same time period. A control group, although not feasible regarding the changes in the workflow and process at the service level, could be included with respect to the SIMC outcomes related to treatment gap, virologic failure, and loss of follow-up. In contrast, this intervention, supported by the STI/AIDS Program from the State of Sao Paulo achieved important results in patients, healthcare services, and program level, when we compared the pre-and post-intervention outcomes. Additionally, we hope that these results will support and improve the reports available in the system. 

The pre/post analyses of responses on the NoMAD survey need further analysis. While the comparison statistics seem appropriate, the very high attrition rate (65%) raises the question of attrition bias in the follow-up sample and challenges confidence in the strength of the findings. This should be addressed by explaining better who did not respond to the follow-up and possible reasons they did not. The current comment, that the non-responders are likely to be service staff not responsible for these tasks, is insufficient. Additionally, this low response rate to the follow-up NoMAD survey should be mentioned in the Limitations section.

Author’s answer: The limitation section was updated to include to approach this limitation about low response rate to NoMAD questionnaire. Changes are available bellow: 

The number of providers that answered the NoMAD questionnaire at the end of the intervention decreased compared with the number at the start of the intervention. This could be related to two factors observed but not measured during the intervention: 1. At the beginning of the intervention, the managers usually asked for all healthcare service providers to answer the questionnaire, even before defining whether all professionals would participate in the intervention. 2. The CCM activities were centered on specific members of each healthcare service and the majority of respondents of the questionnaire at the end of the intervention. 

The protocol has been made available through the Brazilian clinical research registry. However, it is not clear if all data are available to the public.

Author’s answer: The data are available to the public. The status of section data availability was change to “all data are fully available…” and was add the following section on methods: 

“Data were analyzed using the Software Stata/IC 14.0® and are available at http://www.saude.sp.gov.br/centro-de-referencia-e-treinamento-dstaids-sp/publicacoes/publicacoes-download”.

The whole manuscript, including Figures, would benefit from additional proofreading for grammatical and usage issues or missing words. Particular challenges with clarity include the last sentence before Table 6, and sections of the Discussion, especially the 4th, 5th, and 6th paragraphs, among other places here and there throughout the manuscript.

Author’s answer: To attend this solicitation and editor’s recommendation, the manuscript was reviewed by EDITAGE.

Reviewer #2: Thank you for the opportunity to review this manuscript. The authors present an evaluation of a technical support strategy to increase uptake of clinical monitoring of patients with HIV along the treatment cascade in Sao Paulo, Brazil. They used a mixed methods approach, combining pre/post questionnaires with providers, reviews of clinical databases to track patients through the care cascade, and notation of strategies adopted by providers to support uptake of CCM. Questionnaire results suggest that providers are enthusiastic about CCM and plan to sustain it. However, the patient-level analysis is not informative as to the effects of the implementation strategy, as there are no comparators (either pre-implementation or control) against which to judge relative uptake of CCM and its subsequent effects. The paper contains many useful pieces of information but is generally dense and the findings are difficult to interpret. I would recommend the authors consider a quasi-experimental evaluation approach to judge the effects of the technical support implementation strategy on CCM uptake and patient outcomes (e.g., tracking rates of virologic failure among implementing facilities pre- and post-implementation), or reframe this paper as a case study. See below for some major critiques:

- Perhaps most importantly, it is not clear what the data in Tables 2-6 tell us about the effects of the technical support implementation strategy on CCM usage or patient-level cascade outcomes. Table 6 comes closest by comparing patients pre- and post-implementation, but here the focus is on comparing patient characteristics, not patient probability of restarting therapy conditional on being in the pre- or post-implementation period. Useful patient-level estimates of effectiveness would answer questions like, “What proportion of clinic patients are in virologic failure pre-implementation, vs. what proportion are in virologic failure post-implementation?” Ideally, comparisons like this would also include a pre-post comparison with one or more control facilities.

Author’s answer: The tables 2,3 and 5 demonstrated the results from the steps to remove duplicates and ineligible patients from the list of target patients, essentials to understand calculate the reach from REAIM framework. As cited in a previous answer to the other reviewer, these three tables were transformed in just one (Table 2) to improve comprehension. 

Table 4 and 6 demonstrated the profile of the patients assessed by the health care services and helps to understand possible individual factors that could interfere in the results achieved, as discussed in the Discussion section. E.g.: at least 50% of patients with virologic failure did not have adequate % of pills dispensed in the last year, signalizing low adherence. 

Author’s answer: Tables 4 and 6 (renamed to Tables 3 and 4) tell us to much about the individual factors that could affect the intervention results, so we decide to maintain these table just with key patients’ characteristics that could explain the results founded. Additionally, the limitation section was updated to include the limitation related to the lack of a control group. 

- It is unclear how the PRISM framework was incorporated.

Author’s answer: The use of PRIS was clarified in the method section. Changes are available bellow: 

Based on the Pragmatic, Robust Implementation and Sustainability Model (PRISM) for translating research into practice [9], this conceptual framework was applied to identify which recipients could influence the CCM reach, adoption, implementation, maintenance, and effectiveness. Figure 1 shows contextual factors and evaluation of intervention adoption and effectiveness according to the dimensions of the RE-AIM planning and evaluation framework [10]. 

- Several of the RE-AIM components are inappropriately mapped to study indicators. For example, Reach implies an estimate of the number of patients served divided by the number of eligible patients. In this study, your reach indicators appear to be the number of patients still remaining in each cascade bucket (e.g., virologic failure) after data cleaning.

Author’s answer: The RE-AIM indicators were recalculated considering the reviewer comment and the literature. Also, Figure 1 is now presenting the formula used to measure the dimension Reach from RE-AIM for treatment gap, virological failure and LFU and Table 6 synthetize all the results considering the dimensions of this framework. 

- Figure 2 makes it appear that the NoMAD questionnaire was only given at pre-implementation, but Methods Section 3 implies that NoMAD was given pre- and post-implementation to track changes in provider perspectives. Please clarify in Figure 2.

Author’s answer: As recommended, the response to NoMAD was added in the post-implementation step in Figure 2. 

- Regarding the pre/post CCM implementation questionnaire analysis – do you treat the pre- and post- observations as independent, or do you account for observations of the same provider at pre- and post (paired analysis)? This is not clear.

Author’s answer: The analysis considered the sample as independent because this is an implementation science study and assessed real life settings. This information was added in the section 3. Professionals’ perspectives of the implementation. Changes are available bellow: 

To evaluate significant differences from a statistical point of view on professional’s perception regarding the CCM implementation, responses to the items of the questionnaire, for pre- and post-implementation, were compared as independent samples using the Mann–Whitney U test.

- Methods/Population section – were facilities randomly sampled on the basis of patient volume lost to care, or purposively selected? 

Author’s answer: The facilities were selected based on patient’s volume in the SIMC reports. This section was clarified in Population section. Changes are available bellow: 

The services were selected based on the volume of patients lost from the continuum of care in April 2019, consisting of 1,186 patients with treatment gap (11.5% of the State), 1,328 patients with virologic failure (14.8% of the State), and 3,449 patients who were lost to follow-up (12.5% of the State) [3].

---

## [Decision Letter · Decision Letter 1]

10 Mar 2021

PONE-D-20-23813R1

Improving the continuum of care monitoring in Brazilian HIV healthcare services: an implementation science approach

PLOS ONE

Dear Dr. Loch,

Thank you for submitting your manuscript to PLOS ONE. After careful consideration, we feel that it has merit but does not fully meet PLOS ONE’s publication criteria as it currently stands. Therefore, we invite you to submit a revised version of the manuscript that addresses the points raised during the review process.

We look forward to receiving your revised manuscript.

Kind regards,

Petros Isaakidis

Academic Editor

PLOS ONE

Journal Requirements:

Reviewers' comments:

Reviewer's Responses to Questions

**Comments to the Author**

1. If the authors have adequately addressed your comments raised in a previous round of review and you feel that this manuscript is now acceptable for publication, you may indicate that here to bypass the “Comments to the Author” section, enter your conflict of interest statement in the “Confidential to Editor” section, and submit your "Accept" recommendation.

Reviewer #1: (No Response)

Reviewer #2: (No Response)

2. Is the manuscript technically sound, and do the data support the conclusions?

Reviewer #1: Yes

Reviewer #2: Partly

3. Has the statistical analysis been performed appropriately and rigorously? 

Reviewer #1: Yes

Reviewer #2: Yes

4. Have the authors made all data underlying the findings in their manuscript fully available?

Reviewer #1: Yes

Reviewer #2: Yes

5. Is the manuscript presented in an intelligible fashion and written in standard English?

Reviewer #1: No

Reviewer #2: Yes

6. Review Comments to the Author

Reviewer #1: Most of the responses to the previous critiques are thorough and adequate to improve and address problems of the previous manuscript. The paper has been greatly strengthened and clarified, reads much better, provides important information regarding the intervention, and makes an important contribution to implementation science.

A few points remain to be clarified or improved before publication of the paper.

A reviewer requested an explanation of “hybrid type 3 mixed method implementation study.” That term is still not defined, though more detail is provided on the study design. A simple statement that specifically defines “hybrid type 3” would be extremely valuable.

Regarding the concern that there were significantly fewer time-2 survey responses on the NoMAD and implications of this on reported findings in the “Professionals’ Level” section: If it is the case that professionals took the baseline survey who were not intended to participate in the intervention, and only those who received the intervention responded to the follow-up survey, why weren’t the baseline surveys of the unintended for intervention dropped from analysis, and only the baseline and follow-up surveys of those professionals relevant to or receiving the intervention used in these analyses? It seems likely to have affected all of the analyses of statistical significance if people who we not directly related to or involved in the intervention were included in analyses reported in both the text and Tables 5 and 6. The explanation provided in the Discussion section for the reduced follow-up response rate does not change the problem with how the data are reported in the results section.

There still are many grammatical issues with the language. It may need additional editing by a native English speaker. (E.g., see line 314: “…among 234 patients had treatment gap…” among several others throughout the paper.)

Other smaller points:

1. All figures are still fuzzy. Will they be clear in print?

2. It is not clear what the caption to Fig. 1 refers to since it does not seem to correspond to any lettering in the figure and there is no indication of a footnote in the image.

3. Itemized list of ERIC strategies starting line 166-175 – change verbs to “ing” form to follow the phrase “…1, 2 and 3 involved:” e.g., - conducting…; - making… etc. Or else, complete the phrase as a sentence: “…1, 2, and 3 involved the following tasks.”

4. In Figure 3, the lightening bolts are different colors. However, if printed in black and white, this cannot be distinguished. I recommend using different symbols for the different problems (treatment gap, virologic failure, loss to follow-up).

5. The ^ item in Table 2 referenced in the footnote is not evident in the table.

6. Check/correct periods and commas in Table 3 title and row “>80%”. Check elsewhere in the manuscript as well, e.g., line 346.

7. Delete blank column in Figure 7.

Reviewer #2: Thank you for the opportunity to re-review this manuscript. The authors have strengthened the manuscript since the first review round, and the paper is now much clearer. Methodologically, I think the study is still very limited in terms of its ability to assess the effectiveness of the strategies to increase CCM adoption among the health facilities in the sample, given that they had no control group. The authors acknowledge this and other limitations at the end of their discussion section. Nonetheless, the paper presents interesting data that will be useful for colleagues implementing similar interventions.

- Page 12, lines 272-274: Please clarify the denominators used in the proportions here. For example, were 11.5% of the State population of Sao Paulo in treatment gap? Or 11.5% of PLHIV in Sao Paulo who were in the CCM system? Also, should these population totals (e.g., 1186 patients with treatment gap) align with those from the Results section (Page 14 line 300, 1430 patients in treatment gap)? If not, please explain the discrepancy.

7. PLOS authors have the option to publish the peer review history of their article (what does this mean?). If published, this will include your full peer review and any attached files.

Reviewer #1: No

Reviewer #2: No

---

## [Author Response · Author response to Decision Letter 1]

19 Mar 2021

Dear Editor of PLOS ONE,

Thank you for the comments and suggestions made to the manuscript “Improving the Continuum of Care Monitoring in Brazilian HIV Health Care Services: An implementation science approach”, submitted to PLOS ONE. The suggestions and comments contributed to improve our paper with a view to publication in this Journal. Below we present our responses to the comments from editors and reviewers, as well as the changes made, identified in the revised manuscript.

Best regards

Ana Paula Loch

Editors’ comments: 

Reviewer #1: Most of the responses to the previous critiques are thorough and adequate to improve and address problems of the previous manuscript. The paper has been greatly strengthened and clarified, reads much better, provides important information regarding the intervention, and makes an important contribution to implementation science. 

A few points remain to be clarified or improved before publication of the paper. A reviewer requested an explanation of “hybrid type 3 mixed method implementation study.” That term is still not defined, though more detail is provided on the study design. A simple statement that specifically defines “hybrid type 3” would be extremely valuable.

Author’s answer: Thank you. The statement that defines the concept of hybrid type 3 was provided as suggested. 

Regarding the concern that there were significantly fewer time-2 survey responses on the NoMAD and implications of this on reported findings in the “Professionals’ Level” section: If it is the case that professionals took the baseline survey who were not intended to participate in the intervention, and only those who received the intervention responded to the follow-up survey, why weren’t the baseline surveys of the unintended for intervention dropped from analysis, and only the baseline and follow-up surveys of those professionals relevant to or receiving the intervention used in these analyses? It seems likely to have affected all of the analyses of statistical significance if people who we not directly related to or involved in the intervention were included in analyses reported in both the text and Tables 5 and 6. The explanation provided in the Discussion section for the reduced follow-up response rate does not change the problem with how the data are reported in the results section. 

Author’s answer: All the professionals those took survey in baseline had the intention to participate. This study was on real-life and it was conditioning to the local reality, also affected by the pandemic. Additionally, to avoid embarrassment of participants, the survey was done anonymously, which limits the paired comparison. A better explanation was done about this limitation on referred section and on item “Professionals’ perspectives of the implementation”. 

There still are many grammatical issues with the language. It may need additional editing by a native English speaker. (E.g., see line 314: “…among 234 patients had treatment gap…” among several others throughout the paper.)

Author’s answer: The manuscript was reviewed by EDITAGE after we consider all comments from this review. 

Other smaller points:

1. All figures are still fuzzy. Will they be clear in print?

Author’s answer: yes, we printed a version to test and it is clear. 

2. It is not clear what the caption to Fig. 1 refers to since it does not seem to correspond to any lettering in the figure and there is no indication of a footnote in the image.

Author’s answer: The caption referred to the formulas used to calculate the indicator “Proportion of patients in target population reached through this intervention”. Considering this comment, we made available the formulas in the text, removing the caption from the figure 1. Thank you. 

3. Itemized list of ERIC strategies starting line 166-175 – change verbs to “ing” form to follow the phrase “…1, 2 and 3 involved:” e.g., - conducting…; - making… etc. Or else, complete the phrase as a sentence: “…1, 2, and 3 involved the following tasks.”

Author’s answer: The text was changed as suggested. Thank you 

4. In Figure 3, the lightening bolts are different colors. However, if printed in black and white, this cannot be distinguished. I recommend using different symbols for the different problems (treatment gap, virologic failure, loss to follow-up).

Author’s answer: The figure was changed, using different symbols as suggested.

5. The ^ item in Table 2 referenced in the footnote is not evident in the table.

Author’s answer: The item was added in the table as suggested. 

6. Check/correct periods and commas in Table 3 title and row “>80%”. Check elsewhere in the manuscript as well, e.g., line 346.

Author’s answer: The periods and commas were reviewed in all manuscript as suggested. 

7. Delete blank column in Figure 7.

Author’s answer: The blank column was removed as suggested. 

Reviewer #2: Thank you for the opportunity to re-review this manuscript. The authors have strengthened the manuscript since the first review round, and the paper is now much clearer. Methodologically, I think the study is still very limited in terms of its ability to assess the effectiveness of the strategies to increase CCM adoption among the health facilities in the sample, given that they had no control group. The authors acknowledge this and other limitations at the end of their discussion section. Nonetheless, the paper presents interesting data that will be useful for colleagues implementing similar interventions.

- Page 12, lines 272-274: Please clarify the denominators used in the proportions here. For example, were 11.5% of the State population of Sao Paulo in treatment gap? Or 11.5% of PLHIV in Sao Paulo who were in the CCM system?

Author’s answer: The denominators were clarified in this section as suggested. 

Also, should these population totals (e.g., 1186 patients with treatment gap) align with those from the Results section (Page 14 line 300, 1430 patients in treatment gap)? If not, please explain the discrepancy.

Author’s answer: No, this totals are not align because 1186 refers to the beginning of the intervention, but as explained in Figure 2, in the phase 2 we identified new cases in the report, so this number increased. A new sentence was added in this section to clarify this difference.

---

## [Editor Report · Decision Letter 2]

31 Mar 2021

Improving the continuum of care monitoring in Brazilian HIV healthcare services: an implementation science approach

PONE-D-20-23813R2

Dear Dr. Loch,

We’re pleased to inform you that your manuscript has been judged scientifically suitable for publication and will be formally accepted for publication once it meets all outstanding technical requirements.

Kind regards,

Petros Isaakidis

Academic Editor

PLOS ONE
---

## [Editor Report · Acceptance letter]

28 Apr 2021

PONE-D-20-23813R2 

Improving the continuum of care monitoring in Brazilian HIV healthcare services: An implementation science approach 

Dear Dr. Loch:

I'm pleased to inform you that your manuscript has been deemed suitable for publication in PLOS ONE. Congratulations! Your manuscript is now with our production department. 

Kind regards, 

on behalf of

Dr. Petros Isaakidis 

Academic Editor

PLOS ONE